# The Synthesis and Evaluation of Diethyl Benzylphosphonates as Potential Antimicrobial Agents

**DOI:** 10.3390/molecules27206865

**Published:** 2022-10-13

**Authors:** Anna Brodzka, Paweł Kowalczyk, Damian Trzepizur, Dominik Koszelewski, Karol Kramkowski, Mateusz Szymczak, Aleksandra Wypych, Rafał Lizut, Ryszard Ostaszewski

**Affiliations:** 1Institute of Organic Chemistry PAS, Kasprzaka 44/52, 01-224 Warsaw, Poland; 2Department of Animal Nutrition, The Kielanowski Institute of Animal Physiology and Nutrition, Polish Academy of Sciences, Instytucka 3, 05-110 Jabłonna, Poland; 3Department of Physical Chemistry, Medical University of Bialystok, Kilińskiego 1 Str., 15-089 Białystok, Poland; 4Department of Molecular Virology, Institute of Microbiology, Faculty of Biology, University of Warsaw, Miecznikowa 1, 02-096 Warsaw, Poland; 5Centre for Modern Interdisciplinary Technologies, Nicolaus Copernicus University in Toruń, Wileńska 4, 87-100 Toruń, Poland; 6Institute of Mathematics, Informatics and Landscape Architecture, The John Paul II Catholic University of Lublin, 20-708 Lublin, Poland

**Keywords:** benzylphosphonates, antimicrobial activity, Fpg protein-formamidopyrimidine, lipopolysaccharide (LPS)

## Abstract

The impact of substituent at phenyl ring of diethyl benzylphosphonate derivatives on cytotoxic activity was studied. The organophosphonates were obtained based on developed palladium-catalyzed α, β-homodiarylation of vinyl esters protocol. The new synthetic pathway toward 1,2-bis(4-((diethoxyphosphoryl)methyl)phenyl)ethyl acetate was proposed which significantly improves the overall yield of the final product (from 1% to 38%). Several newly synthesized organophosphonates were tested as new potential antimicrobial drugs on model *Escherichia coli* bacterial strains (K12 and R2-R3). All tested compounds show the highest selectivity and activity against K12 and R2 strains. Preliminary cellular studies using MIC and MBC tests and digestion of Fpg after modification of bacterial DNA suggest that selected benzylphosphonate derivatives may have greater potential as antibacterial agents than typically used antibiotics such as ciprofloxacin, bleomycin and cloxacillin. These compounds are highly specific for pathogenic *E. coli* strains based on the model strains used and may be engaged in the future as new substitutes for commonly used antibiotics, which is especially important due to the increasing resistance of bacteria to various drugs and antibiotics.

## 1. Introduction

Organophosphonates are an important class of compounds, as this structural motif is widely present in many pharmaceuticals [1,2], agrochemicals [3] and other biologically active substances [4]. The naturally occurring organophosphonates isolated from various microorganisms exhibits antibacterial properties; this includes fosfomycin (cis-(1R,2S)-1,2-epoxypropylphosphonic acid), 2-amino-5-phosphono-3-pentenoic acid, and 2-amino-4-methylphosphinobutanoic acid (phosphinothricin) [5]. An especially interesting group of compounds are benzylphosphonate derivatives, as they possess important biological activity such as an inhibition of ectonucleosidases (NTPDases) [6], anticancer [7] or antibacterial activity [8] (Figure 1). 

Recently, we have focused our attention on the development of new potent antimicrobial drugs [9,10,11,12,13,14,15,16,17,18] including organophosphonates [17,18]. We have developed several efficient protocols toward the synthesis of this class of compounds, including selective esterification of phosphonic acids [19,20], lipase-catalyzed Kabachnik-Fields [18] and Pudovik reactions [17,21]. We have also studied the biological activity of obtained compounds and proved that various organophosphonates such as α-aminophosphonates [18] or α-acyloxy phosphonates [17] possess antibacterial activity towards different *Escherichia coli* strains. The introduction of phosphate ester group to molecule improves its metabolic stability and membrane transport [21]. Latest reports also indicate that phosphoryl groups located in the benzyl moiety also provide better bioavailability of compounds in biological media [22]. Thus, such benzylphosphonates are good drug candidates. The analysis of literature data indicates that 3,5-difluorobenzylphosphonates (Figure 1) [7] possess antibacterial activity. Unfortunately, to the best of our knowledge, there are no reports concerning the influence of phenyl ring substituents of benzylphosphonate derivatives on their antimicrobial activity. Thus, the aim of this work is the synthesis of various substituted benzylphosphonate diethyl ester derivatives and their validation as antimicrobial agents against model strains of *Escherichia coli* K12 (with native LPS in its structure) and R2-R4 (different lengths of LPS in its structure).

## 2. Results and Discussion

### 2.1. Chemistry

According to literature reports, benzylphosphonate derivatives possess pharmacological activity (Figure 1), thus we would like to study the impact of phenyl ring substituents of benzylphosphonate derivatives on their antimicrobial activity. For this study we prepared various diethyl benzylphosphonate derivatives (**1**–**6**, **8**) and one compound without the presence of a phosphonate group (**7**) (Figure 2). 

Compound **1** was obtained via our previously developed protocol for the selective esterification of phosphonic acids [19]. Diethyl (4-(4,4,5,5-tetramethyl-1,3,2-dioxaborolan-2-yl)benzyl)phosphonate (**2**) and (4-((diethoxyphosphoryl)methyl)phenyl)boronic acid (**3**) were synthesized according to literature protocols [23,24]. The 1,2-bis(4-((diethoxyphosphoryl)methyl)phenyl)ethyl acetate (**4**) can be obtained using known protocols; however, they are non-efficient, and the overall yield of product **4** was 1% (Figure 1A). Thus, we have developed a new approach towards the compound **4** based on our previously developed vinyl esters α,β-homodiarylation method [25], which improved the yield to 38% (Figure 1B). So far, such benzylphosphonate esters were obtained via coupling of *p*-hydroxymethylphenylboronic acid with olefin followed by Arbuzov reaction (Figure 1A), but this approach suffers from low yield. The application of arylboronic acid with phosphonic ester moiety (**3**) as a coupling partner not only improves the reaction efficiency dramatically but also leads to the formation of interesting by-product **8**. Moreover, 4-((diethoxyphosphoryl)methyl)styryl acetate (**5**) and (E)-4,4′-bis(diethylphosphonatemethyl)stilbene (**6**) were obtained in a single reaction vessel as products of Pd-catalyzed coupling of diethyl (4-(4,4,5,5-tetramethyl-1,3,2-dioxaborolan-2-yl)benzyl)phosphonate (**2**) with vinyl acetate.

### 2.2. Cytotoxic Studies of the Library of Synthesized Compounds ***1***–***8***

The synthesis of new compounds from the diethyl benzylphosphonate derivatives group, similarly to other compounds studied by us in previous works, may constitute a new alternative to the commonly used antibiotics in clinical infections. Therefore, the analyses of the obtained compounds are extremely important in nosocomial or clinical infections [26,27,28,29,30,31]. The analyzed compounds were tested for their toxic effect on cells of *Escherichia coli* K12 (having native LPS) and R2-R4 strains (with different LPS lengths). The obtained results show that all tested diethyl esters of benzylphosphonate derivatives **1**–**6**, **8** and 1,2-bis(4-(chloromethyl)phenyl)ethyl acetate (**7**) have a cytotoxic effect on each analyzed *E.coli* bacterial strain differing in LPS length. A different inhibitory activity was noted depending on the nature of the substituent attached to the phenyl ring of organophosphorus compound. Diethyl benzylphosphonate (**1**) exhibits low minimal inhibitory concentration (MIC) (Figure 3). The introduction of pinacol boronic ester group in *para* position of phenyl ring (**2**) did not affect the MIC. Similar results were also obtained for (1,2-bis(4-((diethoxyphosphoryl)methyl)phenyl)ethyl acetate (**4**), 4-((diethoxyphosphoryl)methyl)styryl acetate (**5**) and compound without the presence of phosphonate group **7**. However, boronic acid derivative **3,** (E)-4,4′-bis(diethylphosphonatemethyl)stilbene (**6**) and diethyl ((2′,5′-dioxo-2′,5′-dihydro-[1,1′-biphenyl]-4-yl)methyl)phosphonate (**8**) exhibit increased inhibitory activity towards the strain R2 and similar cytotoxic profile as commonly used antibiotics (Figure 3 and Figure 4 vs. Figure 7). The MIC and MBC test values for each model of *E. coli* R2-R4 and K12 strains were visible in all analyzed growth microplates after the addition of resazurin. 

The MIC denotes the lowest concentration of a biocidal agent (antibiotic or chemotherapeutic) that inhibits the growth of microorganisms (usually referring to bacteria and fungi), most often expressed in mg/L. It is a parameter characterizing, among others, bacteriostatic and bactericidal drugs or substances. The MIC determines which drug or compound concentration inhibits bacterial growth. The MBC denotes the lowest concentration of bactericidal agent (antibiotic) at which 99.9% of bacteria die. It is determined in vitro and expressed in mg/L. MBC is a parameter characterizing antibacterial drugs or compounds; it determines what concentration of the drug has bactericidal activity. Both tests are based on liquid growth media method. 

Model strains of *E.coli* were plotted in all 48-well plates observed; K12, R2-R4 which were treated with the analyzed compounds. From analysis of the MIC and MBC assays, color changes were observed for all compounds tested, but at different levels and at different dilutions. Bacterial strains R3 and R4 were the most susceptible to modification with these compounds due to the increasing length of their LPS (visible dilutions of 10-2 corresponding to a concentration of 0.0015 µM) rather than strains K12 and R2 (visible dilutions of 10-6 corresponding to a concentration of 0.02 µM). The analyzed R4 strain was the most sensitive of all strains, probably due to the longest length of the lipopolysaccharide chain in the bacterial membrane. In all analyzed cases, the MBC values were approximately 60 times higher than the MIC values (Figure 3, Figure 4 and Figure 5 and Table 1).

### 2.3. The Analysis of Bacterial DNA Isolated from E. coli R2–R4 Strains Modified with Tested Biethyl Benzylphosphonate Derivatives 

Based on the observed results in the MIC and MBC tests, it was found that the analyzed compounds significantly influenced the defragmentation of the membrane and the structure of the cell wall of bacteria containing LPS of various lengths [10,11,12,13,14,15,16,17,18,28,29,30,31], causing oxidative stress to damage and modification of bacterial DNA. This was further confirmed by the digestion with a specific enzyme Fpg of modified bacterial DNA (Labjot, New England Biolabs, UK), recognizing oxidized guanine and adenine. The obtained values of oxidative damage, after digestion with the Fpg protein in the bacterial DNA, were compared with the modifications of the bacterial DNA after treatment with antibiotics such as ciprofloxacin, bleomycin and cloxacillin [10,11,12,13,14,15,16,17,18,28,29,30,31] (Figure 6). The presented research shows that diethyl benzylphosphonate derivatives can be used in the future as typical “substitutes” for new drugs in relation to the antibiotics used in hospital infections. The obtained MIC values, as well as our previous studies with various types of the analyzed compounds [10,11,12,13,14,15,16,17,18], indicate that diethyl benzylphosphonate derivatives also show a strong cytotoxic effect on the analyzed model bacterial strains K12 and R2–R4 (Figure 7). Based on the MIC and MBC values, the analyzed compounds **3**, **6** and **8** were selected for further studies (on the basis of their biological highest activity similar to that of antibiotics) on the analysis of oxidative stress in the cell by modifying them with the bacterial DNA. Modified bacterial DNA was digested with Fpg protein from the group of repair glycosylases, which is a marker of oxidative stress [10,11,12,13,14,15,16,17,18]. By using this protein, we wanted to observe whether the resulting modifications in bacterial DNA would introduce oxidative damage to the DNA chain by changing the topological three forms of bacterial DNA: ccc, oc and linear forms, as observed in previous studies [10,11,12,13,14,15,16,17,18]. The results of bacterial DNA modified with the analyzed compounds (Appendix A with the action of Fpg) showed that all analyzed diethyl benzylphosphonate derivatives (Figure 8) with different types of substituents located at phenyl ring can strongly change the topology of bacterial DNA, even after digestion with Fpg protein on the basis of agarose gels. Changes in the main topological forms of the plasmid—ccc, oc and linear—were observed in DNA isolated from model strains and digested with Fpg protein. Approximately 3.5% of oxidative damage was identified after digestion with the Fpg protein, which indicates that the analyzed compounds very strongly damage bacterial DNA as a result of the oxidative stress induced by them in the cell and are highly toxic to it, similarly to the observations of previous studies [10,11,12,13,14,15,16,17,18]. Probably the different types of substituents located at phenyl ring may determine the toxicity of studied compounds on the analyzed *E. coli* strains, including in particular R4, as evidenced by the obtained MIC and MBC values. The obtained results for individual compounds were statistically significant at the level of *p* < 0.05 (Figure 8).

Modifications with antibiotics were smaller and not as clear as in the case of the analyzed compounds (see Appendix A). The sensitivity of *E. coli* strains to the cytotoxic effect of the compounds used and after the digestion of the Fpg protein was as follows: R4 > R2 > R3 > K12 and this effect was very similar to our previous studies [10,11,12,13,14,15,16,17,18,21,22,23,24,25,26,27,28,29,30,31]. This indicates a very high cytotoxicity of the analyzed derivatives towards bacterial DNA, probably resulting from the modification of the components of the bacterial membrane and the LPS contained in it, which may induce specific enzymes from the group of topoisomerases and helicases, destabilizing the structure of the exposed DNA bases. The destabilization of the complex that regulates these enzymes is possibly crucial for the survival of bacterial cells and may play an important role in changing its electrokinetic potential expressing the reversal of burdens. Blocking these enzymes stops DNA replication, causing bacterial cells to apoptosis and be destroyed. In the future, cytotoxicity studies will also be carried out using various cell lines and cultures to assess the biocompatibility of test compounds. Dysfunction of bacterial membranes containing different lengths of LPS in model bacterial strains is an ideal model to assess the effectiveness of these compounds in relation to the antibiotics used [10,11,12,13,14,15,16,17,18,21,22,23,24,25,26,27,28,29,30,31].

Performed studies proved that the analyzed and newly synthesized compounds can potentially be used as “substitutes” for the currently used antibiotics in hospital and clinical infections. 

Large modifications of plasmid DNA were observed for three analyzed compounds, especially for those marked with numbers **3**, **6** and **8** that showed superselectivity in all analyzed bacterial strains, even differentiating the cytotoxicity in the K12 strain. 

## 3. Materials and Methods

### 3.1. Microorganisms and Media

*E. coli* strains R1–R4 and K-12 were obtained as a kind gift from Prof. Jolanta Łukasiewicz at the Ludwik Hirszfeld Institute of Immunology and Experimental Therapy (Polish Academy of Sciences). The reference bacterial strains of *E.coli* (K12 ATCC 25404, R2 ATCC 39544, R3 ATCC 11775, R4 ATCC 39543 were provided from (LGC Standards U.K.) and were used according to the recommendation of ISO 11133: 2014. These strains were used to test antibacterial activity of the synthesized agents [10,11,12,13,14,15,16,17,18,28,29,30,31].

Bacteria were grown in liquid medium or agar plates containing tryptic soy broth medium (TSB; Sigma-Aldrich, Saint Louis, MI, USA), using a methodology described previously [12,13,14,15,16,17,18]. 

### 3.2. Estimation of Minimum Inhibition Concentration (MIC) and Minimum Bactericidal Concentration (MBC)

A drug or compound (visible dilutions of 10-2 corresponding to a concentration of 0.0015 µM) directly kills the vegetative forms of the bacteria. Firstly, the 8 compounds were each diluted in water, at a final concentration of 10 mM (final concentration). Next, 50 μL of all analysed compounds were diluted to 1 mM and plated on a 48-well plate according to the markings. MICs and MBCs were determined using a methodology described previously [12,13,14,15,16,17,18], with serial dilutions performed from 10^−1^ to 10^−7^. Experiments were performed with two independent replicates. An example of the analysis of test compounds of various concentrations using a resusarin dye on microplates (mg·L^−1^) is presented in Appendix A. Based on MIC and MBC values, K12 and other R strains were selected for further investigation. The bacteria were incubated with each of the 8 analysed compounds at a concentration of 0.1 mM for 24 h at 37 °C. Next, the bacterial DNA was isolated from cultures of K12 and R4 *E. coli* using a methodology described previously [12,13,14,15,16,17,18]. 

All microorganisms and media were accurately described in detail in the previous work [12,13,14,15,16,17,18] and analyzed by Tukey test indicated by (*p* < 0.05): * *p* < 0.05, ** *p* < 0.1, *** *p* < 0.01, as described in Table 1.

### 3.3. Chemicals

All the chemicals were obtained from commercial sources and the solvents were of analytical grade. ^1^H and ^13^C NMR spectra were recorded in DMSO-d6, CDCl_3_ or acetone-d6 solution using Bruker Oxford 400 NMR spectrometer (400 MHz). Chemical shifts are expressed in parts per million using TMS as an internal standard. Low-resolution mass spectra were recorded on the API365i API 3000 spectrometer, and the ESI technique was used to analyte ionization. High-resolution mass spectra were recorded on the Maldi SYNAPT G2-S HDMS (Waters, Milford, MA, USA) apparatus with a QqTOF analyser.

### 3.4. The Synthesis of Diethyl Benzylphosphonate *(**1**)*

Diethyl benzylphosphonate was synthesized according to the previously published procedure [19]. Benzylphosphonic acid (1 mmol, 172 mg) and triethyl orthoacetate (30 mmol, 5.5 mL) were mixed overnight at 90 °C. The completion of the reaction was monitored by ^31^P NMR. After the completion of the reaction, an excess of the orthoester was evaporated under reduced pressure. The crude product was purified via a short silica gel column (hexanes/ethyl acetate). The compound **1** was isolated as a transparent oil with 98% yield (0.98 mmol, 224 mg). ^1^H NMR (400 MHz, DMSO-*d*_6_) δ 7.39–7.14 (m, 5H), 3.94 (dq, *J*_H-P_ = 7.9, *J*_H-H_ = 7.0, 4H), 3.21 (d, *J*_H-P_ = 21.6 Hz, 2H), 1.16 (t, *J*_H-H_ = 7.0 Hz, 6H); ^13^C NMR (101 MHz, DMSO-*d*_6_) δ 132.3 (d, *J*_C-P_ = 9.0 Hz), 129.7 (d, *J*_C-P_ = 6.6 Hz), 128.2 (d, *J*_C-P_ = 3.0 Hz), 126.4 (d, *J*_C-P_ = 3.4 Hz), 61.3 (d, *J*_C-P_ = 6.5 Hz), 32.3 (d, *J*_C-P_ = 135.1 Hz), 16.1 (d, *J*_C-P_ = 5.8 Hz); ^31^P NMR (162 MHz, DMSO-*d*_6_) δ 26.5. The NMR data are in accordance with those reported in literature [21].

### 3.5. The Synthesis of Diethyl (4-(4,4,5,5-tetramethyl-1,3,2-dioxaborolan-2-yl)benzyl)phosphonate *(**2**)*

Compound **2** was synthesized following the published procedure [23]. Diethyl 4-brombenzylphosphonic acid (0.65 mmol, 200 mg), bis(pinacolato)diboron (0.78 mmol, 237 mg), and KOAc (2 mmol, 196 mg) were dissolved in dry 1,4-dioxane (7.5 mL) and stirred in a three-neck round-bottom flask equipped with a thermometer under argon for 15 min. The Pd(dppf)Cl2 (0.04 mmol, 29 mg) was added, and the reaction was mixed under Ar at 65 °C overnight. Then, the mixture was cooled to room temperature, filtrated through a short celite-silica gel pad, and washed with ethyl acetate (2 × 25 mL). The filtrate was evaporated, and the crude product was purified by flash chromatography (silica gel, 99:1 to 9:1 DCM/MeOH) to afford colorless oil with 73% yield (0.48 mmol, 168 mg). ^1^H NMR (400 MHz, CDCl_3_) δ 7.77–7.66 (m, 2H), 7.33–7.20 (m, 2H), 4.05–3.88 (m, 4H), 3.15 (d, J_H-P_ = 21.9 Hz, 2H), 1.31 (s, 12H), 1.22–1.17 (m, 6H); ^13^C NMR (101 MHz, CDCl_3_) δ 135.0 (d, *J*_C-P_ = 2.8 Hz), 134.9 (d, *J*_C-P_ = 8.9 Hz), 129.3 (d, *J*_C-P_ = 6.3 Hz), 83.9, 62.3 (d, *J*_C-P_ = 6.5 Hz), 34.1 (d, *J*_C-P_ = 137.9 Hz), 24.9, 16.4 (d, *J*_C-P_ = 5.8 Hz); ^31^P NMR (162 MHz, CDCl_3_) δ 26.2. The NMR data are in accordance with those reported in literature [23].

### 3.6. The Synthesis of (4-((diethoxyphosphoryl)methyl)phenyl)boronic acid *(**3**)*

Compound **3** was synthesized following the published procedure with modifications [24]. To a solution of boronic pinacol ester (0.90 mmol, 320 mg) in acetone (5 mL) and water (5 mL), NH_4_OAc (2 mmol, 155 mg) and NaIO_4_ (2.75 mmol, 590 mg) were added. The reaction mixture was stirred at room temperature for 48 h. Next, it was diluted with water and extracted with ethyl acetate (3 × 15 mL). The organic phase was washed with brine (10 mL), dried over MgSO_4_, filtrated and the solvent volume was reduced to about 5 mL. Finally, the product was precipitated by the addition of cold hexane (30 mL). The product was isolated as a white solid with 83% (0.75 mmol, 203 mg). ^1^H NMR (400 MHz, Acetone-d_6_ + D_2_O) δ 7.80–7.70 (m, 2H), 7.31–7.21 (m, 2H), 4.03–3.91 (m, 4H + D_2_O peak), 3.19 (d, *J*_H-P_ = 21.9 Hz, 2H), 1.16 (t, *J* = 7.1 Hz, 6H); ^13^C NMR (101 MHz, Acetone-*d*_6_ + D_2_O) δ 134.9 (d, *J*_C-P_ = 3.2 Hz), 134.6 (d, *J*_C-P_ = 9.5 Hz), 129.7 (d, *J*_C-P_ = 6.5 Hz), 63.0 (d, *J*_C-P_ = 6.8 Hz), 33.4 (d, *J*_C-P_ = 136.8 Hz), 16.4 (d, *J*_C-P_ = 5.9 Hz); ^31^P NMR (162 MHz, Acetone-*d*_6_ + D_2_O) δ 27.1. The NMR data are in accordance with those reported in literature [24].

### 3.7. The Synthesis of (1,2-bis(4-((diethoxyphosphoryl)methyl)phenyl)ethyl Acetate *(**4**)*

Compound **4** was synthesized based on our previously published procedure [25]. To the mixture of (4-((diethoxyphosphoryl)methyl)phenyl)boronic acid (0.52 mmol, 141 mg), 1,4-benzoquinone (0.37 mmol, 40 mg) and palladium acetate (5 mol%, 3 mg) in water (1.5 mL), vinyl acetate (0.23 mmol, 20 mg) was added and the resulting solution was mixed in a closed vial for 23 h at room temperature (about 23 °C). To terminate the reaction, the mixture was diluted with water (5 mL) and ethyl acetate (5 mL). Then, the phases were separated. After the first separation, the water phase was additionally extracted with two portions of ethyl acetate (2 × 10 mL). In the next step, the combined organic fractions were dried with MgSO_4_ and filtrated through a short celite pad. A portion of silica gel (about 2 g) was added to the collected crude solution and the sample was evaporated to dryness. The dry load prepared in this way was purified via flash chromatography (silica gel, 95:5 to 8:2 hexanes/*i*-PrOH). The product was isolated as colorless semi-solid with a 57% yield (0.13 mmol, 72 mg). ^1^H NMR (400 MHz, Acetone-*d*_6_) δ 7.32–7.24 (m, 4H), 7.24–7.17 (m, 2H), 7.15–7.07 (m, 2H), 5.92 (dd, *J* = 7.9, 6.0 Hz, 1H), 4.04–3.85 (m, 8H), 3.22–3.03 (m, 6H), 1.96 (s, 3H), 1.19 (dt, *J*_H-P_ ≈ *J*_H-H_ = 7.3 Hz, 12H); ^13^C NMR (101 MHz, Acetone-*d*_6_) δ 170.1, 139.8 (d, *J*_C-P_ = 3.6 Hz), 136.7 (d, *J*_C-P_ = 3.6 Hz), 133.0 (d, *J*_C-P_ = 8.8 Hz), 131.5 (d, *J*_C-P_ = 8.9 Hz), 130.7 (d, *J*_C-P_ = 6.6 Hz), 130.6 (d, *J*_C-P_ = 6.6 Hz), 130.3 (d, *J*_C-P_ = 3.2 Hz), 127.4 (d, *J*_C-P_ = 2.9 Hz), 76.9, 62.3 (d, *J*_C-P_ = 6.4 Hz), 43.0, 33.6 (d, *J*_C-P_ = 137.5 Hz), 21.0, 16.7 (d, *J*_C-P_ = 5.7 Hz); ^31^P NMR (162 MHz, Acetone-*d*_6_) δ 25.8, 25.7. HRMS (ESI): *m*/*z* calcd for C_26_H_38_O_8_P_2_+Na^+^: 563.1940 [M + Na]^+^; found: 563.1942.

### 3.8. The Synthesis of 4-((diethoxyphosphoryl)methyl)styryl Acetate *(**5**)*

Compound **5** was obtained via a similar procedure as compound **4**. In this case instead of boronic acid its pinacol ester was used (0.16 mmol, 57 mg) as a substrate, along with vinyl acetate (0.16 mmol, 14 mg), Pd(OAc)_2_ (5 mol%, 1 mg) and 1,4-benzoquinone (0.15 mmol, 16 mg) and water (0.5 mL). The main difference between procedures was the purification process. In this case, the organic fraction after drying and filtering through a celite pad was evaporated to dryness. The obtained crude product was purified via column chromatography (silica gel, 99:1 to 9:1 DCM/MeOH). The product was received as a mixture of *E*/*Z* isomers in a form of pale yellow liquid with 21% yield (0.03 mmol, 10 mg). ^1^H NMR (400 MHz, CDCl_3_) δ 7.81 (d, *J* = 12.8 Hz, 1H), 7.60–7.45 (m, 0.44 H), 7.38–7.03 (m, 8H), 6.36 (d, *J* = 12.8 Hz, 1H), 5.67 (d, *J* = 7.3 Hz, 0.21H), 4.12–3.90 (m, 8H), 3.15 (d, *J* = 21.6 Hz, 1H), 3.13 (d, *J* = 21.7 Hz, 2H), 2.27 (s, 1H), 2.19 (s, 3H), 1.29–1.19 (m, 19H); ^13^C NMR (101 MHz, CDCl_3_) δ 168.9; 167.4; 134.9 (d, *J*_C-P_ = 4.0 Hz), 134.4, (d, *J*_C-P_ = 9.5 Hz), 129.5 (d, *J*_C-P_ = 6.5 Hz), 116.3; 112.3; 61.8 (d, *J*_C-P_ = 6.8 Hz), 33.4 (d, *J*_C-P_ = 136.8 Hz), 21.2; 16.0 (d, *J*_C-P_ = 5.9 Hz); ^31^P NMR (162 MHz, CDCl_3_) isomer E: δ 26.2; isomer Z: δ 26.5. HRMS (ESI): *m*/*z* calcd for C_15_H_21_O_5_P+Na^+^: 335.1024 [M + Na]^+^; found: 335.1027. 

### 3.9. The Synthesis of (E)-4,4′-bis(diethylphosphonatemethyl)stilbene *(**6**)*

Compound **6** was obtained as a second product in the above reaction as a white semi-solid with 16% yield (0.01 mmol, 6 mg). ^1^H NMR (400 MHz, Acetone-d_6_) δ 7.59–7.46 (m, 4H), 7.40–7.29 (m, 4H), 7.22 (s, 2H), 4.07–3.88 (m, 8H), 3.17 (d, *J*_H-P_ = 21.8 Hz, 4H), 1.21 (t, *J* = 7.0 Hz, 12H. The NMR data are in accordance with those reported in literature [26].

### 3.10. The Synthesis of 1,2-bis(4-(chloromethyl)phenyl)ethyl acetate *(**7**)*

Compound **7** was synthesized from 1,2-bis [4-(hydroxymethyl)phenyl]ethyl acetate following the published procedure [27]. Substitution of hydroxyl groups in 1,2-bis [4-(hydroxymethyl)phenyl]ethyl acetate was performed based on the previously published procedure [27]. Substrate (0.48 mmol, 145 mg) was dissolved in dried DCM (5 mL) under argon atmosphere. Then, thionyl chloride was added dropwise in two portions (one portion: 0.97 mmol, 70 μL), and the conversion of the substrate was monitored by TLC (hexanes/EtOAc). After the addition of a second portion, the reaction was stirred under inert gas flow for an additional 2 h, and finally, it was terminated by dilution with a cold 5% NaHCO_3_ solution. The first organic phase was separated, and the water layer was extracted with an additional portion of DCM (10 mL). Combined organic fractions were washed with brine, dried with MgSO_4_, filtered and concentrated on a rotatory evaporator. The crude reaction mixture was purified via flash chromatography (silica gel, 9:1 to 4:6 hexanes/EtOAc) and the product was isolated as the colorless oil with 10% yield (0.05 mmol, 16,2 mg). ^1^H NMR (400 MHz, CDCl_3_) δ 7.38–7.31 (m, 2H), 7.30–7.21 (m, 4H), 7.13–7.04 (m, 2H), 5.93 (dd, *J* = 7.9, 6.0 Hz, 1H), 4.57 (s, 2H), 4.55 (s, 2H), 3.18 (dd, *J* = 13.8, 7.9 Hz, 2H), 3.04 (dd, *J* = 13.8, 6.0 Hz, 2H), 2.02 (s, 3H). ^13^C NMR (101 MHz, CDCl_3_) δ 170.1, 140.3, 137.4, 137.3, 136.0, 130.0, 128.8, 128.7, 127.1, 76.2, 46.2, 46.0, 42.7, 21.3.

### 3.11. The Synthesis of Diethyl ((2′,5′-dioxo-2′,5′-dihydro-[1,1′-biphenyl]-4-yl)methyl)phosphonate *(**8**)*

Compound **8** was isolated as a byproduct of compound **4** synthesis. It was isolated as orange solid (about 1 mg) in an amount insufficient for melting point measurements. ^1^H NMR (400 MHz, Acetone-d_6_) δ 7.59–7.49 (m, 2H), 7.38–7.32 (m, 2H), 6.85–6.77 (m, 2H), 6.66 (dd, *J* = 8.6, 3.0 Hz, 1H), 4.02 (dq, *J*_H-P_ ≈ *J*_H-H_ = 7.2 Hz, 4H), 3.20 (d, *J*_H-P_ = 21.7 Hz, 2H), 1.23 (t, *J* = 7.1 Hz, 6H). ^31^P NMR (162 MHz, Acetone-*d*_6_) δ 26.0. ^13^C NMR (101 MHz, Acetone-*d*_6_) δ 186.6, 185.7, 144.8, 137.8, 137.5, 137.4, 134.3, 134.2, 132.9, 129.1, 129,0, 62.3, 62.2, 33.4, 32.0, 15.7, 15.6. HRMS (ESI): *m*/*z* calcd for C_17_H_19_O_5_P+Na^+^: 357.0868 [M + Na]^+^; found: 357.0873.

## 4. Conclusions

In conclusion, we have synthesized various substituted benzylphosphonate diethyl esters derivatives. Most of them were obtained based on our previously developed palladium-catalyzed α, β-homodiarylation of vinyl esters protocol (compounds **4**–**8**). The presented reactions were carried out in water at ambient temperature without the addition of any ligands, which makes this procedure environmentally benign. We have also proposed the new synthetic pathway toward the coupling reaction products (compound **4**), which significantly improves the overall yield of the final product (from 1% to 38%). The newly synthesized compounds were tested as potential antimicrobial agents and the impact of their structure on the antimicrobial activity against model strains of *Escherichia coli* K12 and R2-R4 were studied. The introduction of boronic acid moiety to the diethyl benzylphosphonate highly improves the antimicrobial activity and selectivity (compound **3** vs. **1**). On the other hand, the antimicrobial activity of the pinacol ester derivative (**2**) is comparable to diethyl benzylphosphonate (**1**). The promising results were also obtained for (E)-4,4′-bis(diethylphosphonatemethyl)stilbene (**6**) and diethyl ((2′,5′-dioxo-2′,5′-dihydro-[1,1′-biphenyl]-4-yl)methyl)phosphonate (**8**).

The tested compounds are able to modify all model strains of *E. coli* (R2-R4) and their bacterial DNA, changing the spatial structure of the LPS contained in their cell membranes. Among the tested phosphonate esters, three compounds **3**, **6**, and **8** showed super-selectivity in all analyzed bacterial strains and exhibited the highest cytotoxic activity, comparable or better than the commonly used antibiotics: ciprofloxacin, bleomycin, and cloxacillin. 

## Data Availability

On request of those interested.

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
