# Peer review of "The Synthesis and Evaluation of Diethyl Benzylphosphonates as Potential Antimicrobial Agents"

_molecules, 2022, doi:10.3390/molecules27206865_

Round 1
Reviewer 1 Report
This manuscript describes the preparation of some phosphonates and their testing as antimicrobial agents.
The manuscript is well written, the topic is interesting and the experimental part is clear and easy to follow.
The purity of the compounds, as judged by the NNMR spectra is good with some impurities only in the first compounds of the synthesis.
There are only a few mistakes that should be cleared easily, such as in line 300, NH3OAc should be changed to NH4OAc, and in line 83 “is suffer” should be changed to “suffers”.
Maybe there is a high degree of auto-citation, with at least 12 of the 31 references belonging to the same group. Since many of them are not directly related to the topic of the present manuscript (the first ones are only related in general to the development of antimicrobial drugs) the number of auto-citations should be reduced to pertinent examples.
Overall, is a good paper that can be published in Molecules.
Author Response
Firstly, we would like to express our gratitude to Reviewers for their suggestions that allowed us to considerably improve our manuscript. To respond to the editor and reviewers queries we introduced a number of changes in the text. We have revised the text according to the suggestions and hope that you will now find it suitable for publication in Molecules journal.
Below, please find the detailed information on the changes in the manuscript with answers to all comments.
Rewiever 1:
- There are only a few mistakes that should be cleared easily, such as in line 300, NH3OAc should be changed to NH4OAc, and in line 83 “is suffer” should be changed to “suffers”..
Answer: The found errors were corrected.
- Maybe there is a high degree of auto-citation, with at least 12 of the 31 references belonging to the same group. Since many of them are not directly related to the topic of the present manuscript (the first ones are only related in general to the development of antimicrobial drugs) the number of auto-citations should be reduced to pertinent examples.
Answer: We are grateful for this opinion. The references 9-16 refer to the development of new antimicrobial drugs (against E. coli), references 17-18 concern organophosphorus compounds as antimicrobial agents, references 19-20, 25 describes the methods for the synthesis of tested compounds.
Reviewer 2 Report
Ostaszewski and coworkers synthesized 7 diethyl benzylphosphonate derivatives and one related compound, and evaluated their antimicrobial agents. Two compounds were identified as antimicrobial agents having comparable activity to ciprofloxacin, bleomycin, and cloxacillin. The development of new antimicrobial agents are important research topic. However, because the 7 phosphonate derivatives were synthesized according the the previously developed methods by authors, improvement on the aspect of chemistry is low. This manuscript should be submitted to more specific journal to medicinal chemistry such as Bioorganic & Medicinal Chemistry Letters.
Author Response
Firstly, we would like to express our gratitude to Reviewers for their suggestions that allowed us to considerably improve our manuscript. To respond to the editor and reviewers queries we introduced a number of changes in the text. We have revised the text according to the suggestions and hope that you will now find it suitable for publication in Molecules journal.
Below, please find the detailed information on the changes in the manuscript with answers to all comments.
Rewiever 2:
- However, because the 7 phosphonate derivatives were synthesized according the previously developed methods by authors, improvement on the aspect of chemistry is low. This manuscript should be submitted to more specific journal to medicinal chemistry such as Bioorganic & Medicinal Chemistry Letters.
Answer: We are grateful for this opinion. However, we the scope of Molecules includes medicinal chemistry. There are several articles published in Molecules presented only biological studies (e.g. Molecules 2022, 27(16), 5194; Molecules 2022, 27(18), 5837; Molecules 2022, 27(12), 3795 etc.). Moreover, the previously developed method has not been applied for the synthesis of phosphonate derivatives and some of them are new compounds.
Reviewer 3 Report
The authors have presented Diethyl Benzylphosphonates as Potential Antimicrobial Agents. The manuscript is within the scope of journal but there are some major concerns with manuscript which needs to be addressed before publication.
1. One of major concern with Organophosphonates is their toxicity and as nerve agents can cause neurotoxicity. What are authors view on this and how the safety of designed compounds is ensured.
2. Abstract: Some results and key findings should be mentioned in the abstract. the abstract should be revised for this and made more informative for the readers.
3. The first paragraph in results and discussion of chemistry section is irrelevant. Either it should be removed or discussed in introduction section describing rationale of designed compounds.
4. Table 1: Synthesis of compound 4. Can author describe how this is represented as table? As per our understanding, its not table but synthetic scheme. Author should take care while indicating such points in manuscript.
5. The yield of compound 8 is 1%? As indicated in table 4 by author. Please explain this.
6. What is relevance of evaluating compound 7 when the study focus on Benzylphosphonates as compound 7 does not come under this class.
7. How the authors have ensured the purity of compounds as some 1H NMR present otherwise and slight impurities can play drastic change in the activity.
8. Section 3.2: The first paragraph of discussion should be given either in discussion part or moved after methodology of determining MIC.
9. Section 3.2: Sentence "A drug or compound of this concentration directly kills the vegetative forms of the bacteria." The author indicate which concentration at start of paragraph?
10. There are multiple abbreviations used in manuscript while the list provided is very small. The author should try to include most of the abbreviations in the list.
11. The manuscript require grammer check. There are few spelling or grammatical mistakes. The manuscript need to be revised for those before final submission. such as
a. Sentence "we have focus our attention..." In intro and chemistry section.
b. "however they are non-efficient". Add coma after word "however"
c. "Thus we have developed a new...". Add coma after word "Thus".
d. "but this approach is suffer from low yield". Correct grammer.
e. Revise "... the diethyl benzylphosphonates derivatives 94 group, similarly..."
and so on........
12. Fig. S23: "modified with selected coumarin derivatives (Panel A) from 10 selected compounds, as shown in" Author indicating which coumarin derivatives? This shows the negligence of authors as it looks copied from some other paper.
Author Response
Firstly, we would like to express our gratitude to Reviewers for their suggestions that allowed us to considerably improve our manuscript. To respond to the editor and reviewers queries we introduced a number of changes in the text. We have revised the text according to the suggestions and hope that you will now find it suitable for publication in Molecules journal.
Below, please find the detailed information on the changes in the manuscript with answers to all comments.
Rewiever 3:
- One of major concern with Organophosphonates is their toxicity and as nerve agents can cause neurotoxicity. What are authors view on this and how the safety of designed compounds is ensured.
Answer: We are very grateful for this comment. We agree the organophosphorus compounds can be toxic and can cause neurotoxicity. However, organosphosponates are of low to moderate toxicity to people. Organophosphates (O=P-(OR)3) which are components of herbicides, insecticides and pesticides are highly toxic (S. X. Naughton, A. V. Terry, Jr.; Toxicology. 2018 Sep 1; 408: 101–112).
- Abstract: Some results and key findings should be mentioned in the abstract. the abstract should be revised for this and made more informative for the readers.
Answer: The abstract was revised to:
“The impact of substituent at phenyl ring of diethyl benzylphosphonate derivatives on cytotoxic activity was studied. The organophosphonates were obtained based on developed palladi-um-catalyzed α, β‐homodiarylation of vinyl esters protocol. The new synthetic pathway to-ward1,2-bis(4-((diethoxyphosphoryl)methyl)phenyl)ethyl acetate was proposed which signifi-cantly improves the overall yield of final product (from 1% to 38%). Several newly synthesized organophosphonates was tested as new potential antimicrobial drugs on model Escherichia coli bacterial strains (K12 and R2-R3). All tested compounds show the highest selectivity and activity against K12 and R2 strains. Preliminary cellular studies using MIC and MBC tests and di-gestion of Fpg after modification of bacterial DNA suggest that selected benzylphosphonate de-rivatives may have greater potential as antibacterial agents than typically used antibiotics, such as ciprofloxacin, bleomycin and cloxacillin . These compounds are highly specific for pathogenic E. coli strains based on the model strains used and may be engaged in the future as new substitutes for commonly used antibiotics what is especially important from the increasing resistance of bacteria to various drugs and antibiotics point of view.”
- The first paragraph in results and discussion of chemistry section is irrelevant. Either it should be removed or discussed in introduction section describing rationale of designed compounds.
Answer: This paragraph was transferred to the introduction section. “[…]Recently, we have focused our attention on the development of new potent antimicrobial drugs [9-18] including organophosphonates [17-18]. We have developed several efficient protocols toward the synthesis of this class of compounds including, selective esterification of phosphonic acids [20], lipase-catalyzed Kabachnik-Fields [18] and Pudovik reactions [17, 21]. We have also studied the biological activity of obtained compounds and proved that various organophosphonates such as a-aminophosphonates [18] or a-acyloxy phosphonates [17] possess antibacterial activity towards different Escherichia coli strains. The introduction of phosphate ester group to molecule improves its metabolic stability and membrane transport. [21].”
- Table 1: Synthesis of compound 4. Can author describe how this is represented as table? As per our understanding, its not table but synthetic scheme. Author should take care while indicating such points in manuscript.
Answer: There is an error in manuscript. The Table 1 should be titled as Scheme. This error was corrected.
- The yield of compound 8 is 1%? As indicated in table 4 by author. Please explain this.
Answer: The compound 8 was obtained as a by-product with 1% of yield. The yield was calculated for the amount of phenylboronic acid. We have performed the mentioned reaction several times to obtain total amount of compound 8 sufficient for NMR analysis and biological studies.
- What is relevance of evaluating compound 7 when the study focus on Benzylphosphonates as compound 7 does not come under this class.
Answer: The aim of evaluation of compound 7 was to show the impact of phosphonate group on the antimicrobial activity.
- How the authors have ensured the purity of compounds as some 1H NMR present otherwise and slight impurities can play drastic change in the activity.
Answer: The impurities presented in some 1H NMR spectra are residual solvents that were not removed by drying in vacuo. However, the HNMR spectra indicate that the purity of compounds is above 95%. We have also performed the biological studies for dried compound 1 (the compound was dried in vacuo overnight) obtaining the same results as previously.
- Section 3.2: The first paragraph of discussion should be given either in discussion part or moved after methodology of determining MIC.
Answer: Point 3.2: The first paragraph of the discussion has been moved from the methodological description to chapter 2.2. in the discussion
- Section 3.2: Sentence "A drug or compound of this concentration directly kills the vegetative forms of the bacteria." The author indicate which concentration at start of paragraph?
Answer: The missing brace was inserted at the beginning of the paragraph
- There are multiple abbreviations used in manuscript while the list provided is very small. The author should try to include most of the abbreviations in the list.
Answer: Abbreviations have been included in greater numbers and the abbreviations list was corrected.
- The manuscript require grammer check. There are few spelling or grammatical mistakes. The manuscript need to be revised for those before final submission. such as
- Sentence "we have focus our attention..." In intro and chemistry section.
- "however they are non-efficient". Add coma after word "however"
- "Thus we have developed a new...". Add coma after word "Thus".
- "but this approach is suffer from low yield". Correct grammer.
- Revise "... the diethyl benzylphosphonates derivatives 94 group, similarly..."
and so on........
Answer: The found errors was corrected.
- S23: "modified with selected coumarin derivatives (Panel A) from 10 selected compounds, as shown in" Author indicating which coumarin derivatives? This shows the negligence of authors as it looks copied from some other paper.
Answer: There was an error in manuscript and it was corrected.
Round 2
Reviewer 2 Report
This manuscript should be published in more specific journal of medicinal chemistry.
Reviewer 3 Report
NA